# Pen Versus Crate: A Comparative Study on the Effects of Different Farrowing Systems on Farrowing Performance, Colostrum Yield and Piglet Preweaning Mortality in Sows under Tropical Conditions

**DOI:** 10.3390/ani13020233

**Published:** 2023-01-08

**Authors:** Natchanon Dumniem, Rafa Boonprakob, Thomas D. Parsons, Padet Tummaruk

**Affiliations:** 1Department of Obstetrics, Gynaecology and Reproduction, Faculty of Veterinary Science, Chulalongkorn University, Bangkok 10330, Thailand; 2School of Veterinary Medicine, University of Pennsylvania, Kennett Square, Philadelphia, PA 19348-1692, USA; 3Center of Excellence in Swine Reproduction, Chulalongkorn University, Bangkok 10330, Thailand

**Keywords:** animal welfare, backfat thickness, colostrum, farrowing, piglet preweaning mortality

## Abstract

**Simple Summary:**

Loose-housed pens are being implemented as alternative farrowing systems in the swine industry worldwide. This system allows sows to express natural behaviour and reduces stress during the peripartum period. However, most intensive swine farms in Thailand still confine sows in crates during lactation to minimise piglet mortality due to crushing. The present study was performed to compare the reproductive performance of sows kept in the farrowing crate and in the free-farrowing system under tropical conditions. Sows kept in the free-farrowing system produced more colostrum than crated sows. Piglet preweaning mortality rate and the proportion of piglet loss due to crushing in free-farrowing sows were greater than in crated sows. Sow farrowing performance, newborn piglet characteristics and milk production did not differ between the two farrowing systems. Interestingly, in the free-farrowing system, the incidence of crushing in sows with high backfat thickness was significantly higher than in those with moderate and low backfat thickness. These findings imply that free-farrowing pens can be applied in tropical environments without impairing sow farrowing and can enhance sow colostrum production. However, intensive management strategies should focus on adjusting the body conditions of sows prior to farrowing to avoid crushing piglets.

**Abstract:**

The present study was performed to determine the farrowing performance of sows, newborn piglet characteristics, colostrum yield, milk yield and piglet preweaning mortality in a free-farrowing pen compared to a conventional farrowing crate system in a tropical environment. A total of 92 sows and 1344 piglets were included in the study. The sows were allocated by parity into two farrowing systems, either a free-farrowing pen (*n* = 54 sows and 805 piglets) or a crate (*n* = 38 sows and 539 piglets). Backfat thickness and loin muscle depth of sows at 109.0 ± 3.0 days of gestation were measured. Reproductive performance data including total number of piglets born (TB), number of piglets born alive (BA), percentage of stillborn piglets (SB) and percentage of mummified foetuses (MF) per litter, farrowing duration, piglet expulsion interval, time from onset of farrowing to the last placental expulsion, piglet preweaning mortality rate, percentage of piglets crushed by sows and number of piglets at weaning were analysed. In addition, piglet colostrum intake, colostrum yield, Brix index and milk yield of sows were evaluated. On average, TB, BA, farrowing duration, colostrum yield and milk yield during 3 to 10 and 10 to 17 days of lactation were 14.7 ± 2.8, 12.8 ± 3.1, 213.2 ± 142.2 min, 5.3 ± 1.4 kg, 8.6 ± 1.5 kg, and 10.4 ± 2.2 kg, respectively. Sows kept in the free-farrowing pen tended to produce more colostrum than crated sows (5.5 ± 0.2 vs. 4.9 ± 0.2 kg, *p* = 0.080). Piglets born in the free-farrowing pen had a higher colostrum intake than those in the crate system (437.0 ± 6.9 and 411.7 ± 8.3 g, *p* = 0.019). However, the piglet preweaning mortality rate (26.8 ± 2.9 vs. 17.0 ± 3.8, *p* = 0.045) and the proportion of piglets crushed by sows (13.1 ± 2.1 vs. 5.8 ± 2.7, *p* = 0.037) in the free-farrowing pen were higher than those in the crate system. Interestingly, in the free-farrowing pen, piglet preweaning mortality rate in sows with high backfat thickness was higher than that in sows with moderate (37.8 ± 5.1% vs. 21.6 ± 3.6%, *p* = 0.011) and low (21.0 ± 6.2%, *p* = 0.038) backfat thickness. Moreover, the incidence of crushing in sows with high backfat thickness was higher in the free-farrowing pen than in the crate system (17.6 ± 3.6 vs. 4.0 ± 5.7, *p* = 0.049), but this difference was not detected for sows with moderate and low backfat thickness (*p* > 0.05). Milk yield of sows during 3 to 10 days (8.6 ± 0.2 vs. 8.6 ± 2.3, *p* > 0.05) and 10 to 17 days (10.2 ± 0.3 vs. 10.4 ± 0.4, *p* > 0.05) did not differ between the two farrowing systems. In conclusion, piglets born in the free-farrowing pen had a higher colostrum intake than those in the crate system. However, the piglet preweaning mortality rate and the proportion of piglets crushed by sows in the free-farrowing pen were higher than in the crate system. Interestingly, a high proportion of piglet preweaning mortality in the free-farrowing system was detected only in sows with high backfat thickness before farrowing but not in those with low and moderate backfat thickness. Therefore, additional management in sows with high backfat thickness (>24 mm) before farrowing should be considered to avoid the crushing of piglets by sows.

## 1. Introduction

In recent decades, animal welfare has become an issue of interest in the intensive swine industry. In many European countries, the use of gestation crates has been limited or prohibited in most periods of pregnancy except for the first month of gestation and the week before farrowing [1]. The gestation crate fails to meet all of a sow’s biological requirements in part by limiting her ability to perform several natural behaviours, including simply turning around. These altered behavioural responses result from the central nervous system processing of both internal and external stimuli and can frustrate a sow, evoking negative emotional responses and potentially compromising her well-being [2]. The farrowing crate places similar limitations on the periparturient sow. Compromised behavioural responses of sows prior to farrowing is associated with untoward physiological and/or endocrine responses during parturition and lactation periods.

Norway, Finland, Sweden and Switzerland have banned the use of farrowing crates and replaced them with pen-based farrowing systems for lactating sows [1,3]. These so-called free-farrowing systems feature a loose-housed design that allows sows to move freely during the transition and lactation periods and are designed to be an alternative to conventional crated farrowing systems [4]. Sows in free-farrowing pens bedded with straw have lower cortisol responses to the corticotropic-releasing hormone challenge test than crated sows, indicated a lower stress response [5]. Moreover, penned sows tended to have a higher oxytocin pulse than crated sows, which benefits the farrowing process [6]. These findings provide evidence for possible animal welfare benefits to sows housed in free-farrowing systems.

Litter size at farrowing, which has been dramatically increased in modern swine genetics via selective breeding [7], portends several challenges for sows farrowing in either conventional or alternative facilities. This includes prolonged farrowing duration, a variation in newborn piglet birthweight, insufficient colostrum intake and an increased piglet preweaning mortality rate [7]. A previous study over almost three decades revealed the trend of increasing litter size and farrowing duration in modern hyperprolific sows [7]. The process of delivering foetuses causes visceral pain, and its magnitude will be proportional to the number of offspring and the length of parturition [8]. Prolonged farrowing impairs placenta expulsion and increases the risks of postpartum metritis and retained placenta [7]. Continual uterine contraction can cause umbilical cord rupture, meconium staining and peripartum death of piglets [9]. These findings indicate that a long farrowing duration compromises sow welfare as well as health in postpartum and lactation periods. Postpartum complications due to prolonged farrowing duration are exacerbated by barren farrowing environments, for example, confinement or the lack of nest building material [10].

Another consequence of large litter size is that a certain proportion of newborn piglets suffer from intrauterine growth restriction (IUGR). This results in high variation in piglet birthweight within the litter; high competition for colostrum intake, often compromising their ability to achieve sufficient colostrum consumption; and the increased number of low-viability piglets. Large litter size also results in an increased frequency of sows experiencing a negative energy balance during lactation due to their need to produce large volumes of milk [11,12]. Backfat thickness and loin muscle depth are the body-condition parameters associated with sow feed intake, milk yield and lactation performance [11,12,13,14]. Sows with high backfat thickness before farrowing have an increased farrowing duration and piglet expulsion interval [15], leading to a high backfat loss during lactation [13]. However, backfat thickness and loin muscle depth at farrowing are positively correlated with both sow milk yield [13] and milk fat content during lactation [11].

Increased litter size also has begotten increased preweaning mortality. In Thailand, the piglet preweaning mortality average is 11.2% and varies from 4.8% to 19.2% among herds [16], with 78.5% of preweaning mortality occurring within the first 72 h postpartum [17]. Even a short period of peri-parturient asphyxia and hypoxia can lead to brain damage, increase the piglet’s risk of being crushed by a sow and compromise piglet vitality during early postnatal life [18]. The distinct elevation of piglet preweaning mortality rate has become a both a production and welfare concern within the modern swine industry, especially in free-farrowing systems [19]. In the loose-housed system, the primary cause of piglet mortality based on post-mortem examination is trauma, which is most likely associated with crushing by sows [20].

Taken together, these challenges highlight the opportunities for improvement in managing the modern hyperprolific sow of today. Much less is known about how free-farrowing systems impact these factors and to our knowledge have never been examined in a tropical environment. They are all important issues to be considered in the design and adoption of pen-based farrowing systems. Thus, before the further implementation of loose-housed farrowing pens in the large-scale swine industry under tropical conditions, additional knowledge associated with both sow health and piglet characteristics is required. Hence, the present study determined the farrowing performance of sows including the dynamics of backfat thickness and its role on newborn piglet characteristics, colostrum yield, milk yield and piglet preweaning mortality in a free-farrowing system compared to a crated system in a tropical environment.

## 2. Materials and Methods

### 2.1. Animals and Experimental Design

The experiment was conducted in a commercial breeding farm with a herd size of 5000 sows, located in central Thailand, in May to August 2022. A total of 101 crossbred sows (Canadian Landrace × Yorkshire) were randomly allotted to one of two farrowing systems, (i) farrowing crates (*n* = 45) and (ii) free-farrowing pens (*n* = 56), from entering the farrowing unit until weaning. Parity number of sows averaged 2.1 ± 0.6 (range 1 to 3). The experiment was carried out from 7 days before parturition until weaning in two consecutive replicates. The average lactation length was 22.1 ± 0.9 days (range 21 to 24 days). For multiparous sows, the type of the farrowing structures that the sows had previously experienced was the conventional crate system. The farrowing processes of the sows were monitored closely from the start to the end by the research team. Data on sow farrowing characteristics, piglet birthweight, body weight at 24 h postpartum and piglet preweaning mortality were collected. Piglet colostrum intake, sow colostrum yield and sow milk yield were determined. Sow colostrum IgG was estimated by using the Brix refractometer [21]. The experiment was reviewed and approved by the Institutional Animal Care and Use Committee (IACUC) in accordance with the university regulations and policies governing the care and use of experimental animals (protocol number 2131053).

### 2.2. Housing and General Management

Pregnant gilts and sows that were raised in a group-housed system with 280 females per pen, equipped with six electronic sow feeders, were included in the experiment. Gilts and sows were moved to the indoor farrowing house with an evaporative cooling system and temperature control facilities 7 days before the expected parturition date (109 ± 3 days of gestation). After entering the farrowing house, the sows were randomly divided into two groups: farrowing crate and free-farrowing pen. The farrowing pen was designed with an adjustable metal swing hinge and a fully plastic slatted floor, measuring 2.00 × 2.35 × 0.90 m and providing the total area of 4.7 m^2^ per pen. In the farrowing crate system, the metal swing hinge was permanently closed, and the sows were kept in individual crates (1.80 × 0.60 × 0.90 m) with a space allowance of 1.08 m^2^ per sow. In the free-farrowing system, to create a loose farrowing environment, the swing hinge was completely opened, providing a space allowance of 3.25 m^2^ per sow during the whole experimental period. The space allowance for sows in the free-farrowing system in Thailand was designed following the criteria of the minimum space requirement for sows that are able to turn around in their nest space for piglet inspection and gathering behaviour (i.e., 3.17 m^2^) in the farrowing pen [22,23]. In the creep area, a heating lamp, a rubber mattress and a feeding bowl were installed. The schematic diagram of the farrowing pen design is illustrated in Figure 1. This farrowing pen design has been used as an alternative farrowing system in a commercial swine herd in Thailand for over 2 years [24]. Gilts and sows were fed with a commercial lactation diet (907 BTG, Betagro Public Co., Ltd., Lopburi, Thailand) via an automatic feeding pipeline, with an averaged feed allowance of 3.0 to 3.5 kg/sow/day before farrowing, and the feed was provided to ad libitum from parturition date until weaning to meet or exceed their nutritional requirements. The lactation diet contained 13.1% crude protein, 3.68 Mcal/kg metabolisable energy and 0.8% lysine. Drinking water was provided ad libitum via nipples for sows and piglets. After parturition, sows were intramuscularly administered an anti-inflammatory drug (6 mg/kg of ketoprofen, Bezter Ketofen Tec 100^®^, Siam Bioscience Co., Ltd., Nonthaburi, Thailand) and an antibiotic drug (10 mg/kg of amoxicillin, Vetrimoxin L.A.^®^, Ceva Santé Animale, Libourne, France). The creep feed was provided for all litters from 3 days postpartum onwards. Piglet general husbandry included iron injection (200 mg/piglet of iron dextran, Bezter Irondex 100^®^, Thainaoka Pharmaceutical Co., Ltd., Samut Sakhon, Thailand), and teeth clipping was carried out at 1 day of age. Additionally, antiprotozoal drug provision (20 mg/kg of 5% toltrazuril, Better Pharma Co., Ltd., Lopburi, Thailand) and castration were performed at 3 days of age. Cross-fostering was performed to balance the sow functional teats and the number of nursing piglets in the same treatment within 2 days postpartum. Routine sow health care and the vaccination programme were handled by a veterinarian. All gilts and sows were vaccinated against foot and mouth disease virus, porcine circovirus, classical swine fever virus, pseudorabies virus, porcine reproductive and respiratory virus and porcine parvovirus. All sows were kept in a close-housed system equipped with an evaporative cooling system and temperature control facilities (DOL-532, SKOV A/S, Roslev, Denmark) to maintain an optimal temperature inside the barn. The average indoor temperature and humidity during the experimental period were 28.1 ± 1.5 °C (range 25.0 to 32.4 °C) and 74 ± 5.4% (range 67% to 91%), respectively. The proportion of days when the average temperature inside the barn rose above 25.0 °C during the experimental period was 97.4%. In addition, the average maximum daily temperature inside the barn during the experimental period was 30.7 ± 0.9 °C (range of 28.9 to 32.4 °C).

### 2.3. Farrowing Supervision and Characteristics

The farrowing process was carefully supervised by the research team for 24 h a day. Farrowing induction was not applied in this study. Farrowing assistance was performed only when dystocia was clearly identified. Sow dystocia was defined when an interval of over 45 min elapsed from the birth of the previous piglet and the sow showed intermittent straining, accompanied by the paddling of her legs without any piglet being delivered. Birth assistance included the manual extraction of the piglet and the intramuscular administration of oxytocin (20 IU/sow, Oxytocin Synth, Kela N.V., Hoogstraten, Belgium). The newborn piglet was grabbed immediately after birth and evaluated for the meconium staining score according to Mota-Rojas et al. [9]. Thereafter, the piglet was gently rubbed with a dry towel to remove the remaining amniotic sac, the umbilical cord was cut and tied with a sterilised thread, and the piglet was covered with hygienic powder (Farmasec, Farmapro, Plestan, France). All liveborn, stillborn and mummified foetuses were counted and numbered to determine the birth order. Subsequently, individual liveborn piglets were weighed using a digital scale (SDS^®^ IDS701–CSERIES, SDS Digital Scale Co. Ltd., Yangzhou, China). Intrauterine growth restriction (IUGR) was scored according to Bahnsen et al. [25]. Briefly, the piglets were classified as ‘0’ when their physiological appearance was normal (i.e., normal head shape). The piglets were defined as ‘1’ when they experienced mild IUGR (i.e., steep or dolphin-like forehead, narrow hind part, with a maximum of one secondary parameter) and ‘2’ when they experienced severe IUGR (i.e., steep or dolphin-like forehead, distinctively narrow hind part, with at least one secondary parameter) [25]. The secondary parameters included bulging eyes, wrinkles perpendicular to the mouth, spiky hair and unstable mobility. All neonatal management procedures were done within 3 min after delivery. Farrowing duration was defined as the time interval between the delivery of the first and last piglets. The piglet expulsion interval was defined as the time between the births of two consecutive piglets. The cumulative expulsion interval was defined as the difference in the time between the delivery of the first piglet and the time noted for piglet delivery within the same sow. Time from onset of farrowing to the last placental expulsion was defined as the time between the delivery of the first piglet and the expulsion of the last compartment of the placenta. The coefficient of variance (CV) of piglet birthweight was calculated for each litter.

### 2.4. Sow Measurement and Data Collection

Sow identities, parity number, insemination date, farrowing date, weaning date, total number of piglets born (TB), number of piglets born alive (BA), number of stillborn piglets, number of mummified foetuses per litter, number of nursed and weaned piglets and weaning-to-service interval were recorded. The percentages of stillborn piglets (SB) and mummified foetuses (MF) per litter were calculated by dividing the number of stillborn piglets or number mummified foetuses per litter with TB and multiplying it by 100. All gilts and sows were evaluated for backfat thickness and loin muscle depth twice at entering the farrowing house and at 21 days of lactation, using a linear array probe and a real-time B mode ultrasonography (HS–2200, Honda Electronics Co., Ltd., Toyohashi, Aichi, Japan). To measure backfat thickness and loin muscle depth, the ultrasound probe was placed approximately 6.5 cm from the dorsal midline at the last rib curve. Lactational backfat loss was calculated by dividing the difference between backfat thickness at entering the farrowing house and at 21 days of lactation with backfat thickness at entering the farrowing house multiplied by 100. Likewise, lactational loin muscle loss was calculated by dividing the difference between loin muscle depth at entering the farrowing house and at 21 days of lactation with loin muscle depth at entering the farrowing house multiplied by 100.

### 2.5. Piglet Measurement and Preweaning Mortality Data

All live piglets were weighed individually at birth and 24 h after birth. Piglet weight gain at 1 day old was calculated and used to estimate piglet colostrum intake [26]. During lactation, the piglets were weighed at 3, 10, 17 and 21 days of life. Litter weight was calculated by summing all individual piglet body weights. The date and cause of death were recorded for all dead piglets from 1 to 21 days of lactation. Post-mortem examination was not performed because of the farm disease-control policies. However, the cause of death was determined by the observation of the external lesions of the piglet. Piglet preweaning mortality was classified as ‘crush’ if the piglet presented external traumas or lacerations or fractures of major bones, ‘weak’ if the piglet was dead with low birthweight and no external lesions were found and ‘miscellaneous’ if the piglet was dead from other causes not mentioned above. Dead piglets were noted on a daily basis. Piglet preweaning mortality was considered for two periods, including early mortality (the first 3 days of postnatal life) and late mortality (from 4 to 21 days of postnatal life). The piglet preweaning mortality rate of each period was calculated by dividing the total number of dead piglets in the timeframe with BA and multiplying it by 100. Likewise, the proportion of piglets crushed by sows in each period was calculated by dividing the total number of crushed piglets by BA and multiplying it by 100. The piglet preweaning mortality rate during lactation was derived from the piglet preweaning mortality rate of early and late mortality.

### 2.6. Colostrum and Milk

The colostrum intake of individual piglets was calculated using the equation reported by Thiel et al. [26]: −106 + 2.26 WG + 200 BWB + 0.111 D − 1414 WG/D + 0.0182 WG/BWB. Sow colostrum yield was calculated by summing the colostrum intake of all piglets within the litter. Milk yield was estimated using the equation reported by Hansen et al. [27]: milk yield day 3 to 10 (g) = 1.93 + 0.07 × (litter size − 9.5) + 0.04 × (litter gain, kg/day − 2.05). Milk yield day 10 to 17 (g) = 2.23 + 0.05 × (litter size − 9.5) + 0.23 × (litter gain, kg/day − 2.05). Furthermore, within 1 h after the onset of parturition, the Brix refractometer (Pocket PAL–1 refractometer, Atago, Tokyo, Japan) was used to estimate the colostrum IgG [21]. The colostrum sample (0.3 mL) was collected manually from the first three pair of teats of the sows and was dropped into the prism chamber of the Brix refractometer using a disposable plastic dropper. The Brix index value was determined immediately after testing.

### 2.7. Statistical Analysis

All analyses were performed using the statistical analysis system (SAS) software version 9.4 (SAS Institute Inc., Cary, NC, USA). Of the 101 sows, data of sows with litter size less than 8 (*n* = 8) and incomplete farrowing supervision (*n* = 1) were excluded from the analyses. Based on these exclusion criteria, 7 sows in the crate system and 2 sows in the free-farrowing system were excluded, leaving 92 sows and 1344 piglets for data analyses. Descriptive statistics on reproductive data were determined using the MEAN and FREQ procedures of SAS. To differentiate sow lipid deposition, backfat thickness prior to parturition was classified as low (<18 mm), moderate (18 to 24 mm) or high (>24 mm). Continuous data of sows including gestation length, TB, BA, SB, MF, farrowing duration, time from farrowing onset to the last placental expulsion, colostrum yield, Brix index, milk yield from 3 to 10 days and 10 to 17 days of lactation, CV of piglet birthweight within the litter, piglet preweaning mortality rate, proportion of piglets crushed by sow, number of weaned piglets and weaning-to-service interval were analysed by the general linear model (GLM) procedure of SAS. The factors included in the statistical models included farrowing systems (farrowing crate and free-farrowing pen), classes of backfat thickness prior to parturition (low, moderate and high) and their interaction. Least square means of each class of variables were compared using the Tukey–Kramer test. Moreover, sow metabolic parameters, including backfat thickness prior to parturition and at 21 days of lactation, loin muscle depth prior to parturition and at 21 days of lactation and lactational backfat thickness and loin muscle depth loss, were analysed using the GLM procedure of SAS. Piglet characteristics including individual piglet birthweight, piglet expulsion interval, cumulative expulsion interval and colostrum intake were analysed by the general linear mixed model (MIXED) procedure of SAS. The statistical models included the farrowing system (crate and pen), classes of backfat thickness prior to parturition (low, moderate and high) and their interaction as a fixed effect. Sow identities were included in the statistical models to adjust for repeated measurement of the piglet parameters for each sow. Least square means in each class of the variables were compared by using the Tukey–Kramer test. According to Tummaruk and Sang-Gassanee [28], a farrowing duration exceeding 240 min was considered a prolonged farrowing duration. The proportion of sows that had a prolonged farrowing duration (>240 min) for the crate and free-farrowing systems was compared using Chi-square tests. Additionally, the proportions of meconium-stained piglets (score 0 vs. score 1 and 2) and IUGR piglets (score 0 vs. score 1 and 2) were compared between farrowing systems by using Chi-square tests. For all analyses, a *p* value below 0.05 was considered statistically significant, and a *p* value between 0.05 and 0.10 indicated a tendency.

## 3. Results

Across groups, the average TB, BA, SB and MF levels were 14.7 ± 2.8, 12.8 ± 3.1, 9.2% and 3.7%, respectively. Furthermore, farrowing duration, colostrum yield, Brix index and milk yield during 3 to 10 and 10 to 17 days of lactation (means ± SD) were 213.2 ± 142.2 min, 5.3 ± 1.4 kg, 25.7 ± 3.4%, 8.6 ± 1.5 kg and 10.4 ± 2.2 kg, respectively.

### 3.1. Sow Characteristics

#### 3.1.1. Gestation Length, Litter Traits and Sow Metabolic Parameters

Gestation length did not differ between sows kept in the farrowing pen compared with those in the farrowing crate (114.4 ± 0.3 vs. 114.8 ± 0.2 days, *p* > 0.05). Litter traits and metabolic parameters of sows in the free-farrowing system compared to those in the crate system are presented in Table 1, and the different classes of backfat thickness prior to parturition are presented in Table 2. Sows with low backfat thickness prior to parturition had a TB 1.7 higher than that of sows with moderate backfat thickness (*p* = 0.025, Table 2). However, backfat thickness and loin muscle depth prior to parturition were not different between the farrowing systems (*p* > 0.05). Furthermore, sows with low backfat thickness prior to parturition lost less backfat during lactation than those with moderate (16.5 ± 3.1% vs. 28.3 ± 1.9%, *p* = 0.002) and high backfat thickness (33.4 ± 3.2%, *p* < 0.001).

#### 3.1.2. Farrowing Performance

Farrowing duration and time from the onset of farrowing to the last placental expulsion of sows did not differ between crate and free-farrowing systems (Table 3). Besides, farrowing duration did not differ among those with low, moderate or high backfat thickness prior to parturition (*p* > 0.05). The proportion of sows that had a prolonged farrowing duration was similar for the two systems (*p* > 0.05). Regarding piglet traits, piglet expulsion interval, cumulative expulsion interval, individual piglet birthweight and proportion of low-body-weight piglets (<1.0 kg), the proportion of meconium-stained piglets and IUGR piglets were not different between the two farrowing systems (*p* > 0.05) (Table 3).

#### 3.1.3. Colostrum Yield, Milk Yield and Brix Index

Sows kept in the free-farrowing system tended to produce more colostrum than confined sows (*p* = 0.080, Figure 2a). However, the Brix index did not differ between sows kept in the crate and those in the free-farrowing systems (25.7 ± 0.7 vs. 25.6 ± 0.5, *p* > 0.05). Regardless of the farrowing system, sows with high backfat thickness prior to parturition had a higher colostrum yield than those with low backfat thickness (5.7 ± 0.3 vs. 4.8 ± 0.3 kg, *p* = 0.065). Milk yield of sows during 3 to 10 days (8.6 ± 0.2 vs. 8.6 ± 2.3 kg, *p* > 0.05) and 10 to 17 days (10.2 ± 0.3 vs. 10.4 ± 0.4 kg, *p* > 0.05) of lactation did not differ between the two farrowing systems. In addition, in high-backfat sows, milk production during 3 to 10 days (8.3 ± 0.4 vs. 9.6 ± 0.6 kg, *p* = 0.059) and 10 to 17 days (9.6 ± 0.6 vs. 11.7 ± 0.9 kg, *p* = 0.050) of lactation were lower in the free-farrowing system compared to the crate system.

### 3.2. Piglet Characteristics

#### 3.2.1. Piglet Measurement and Colostrum Intake

Individual piglet birthweight, CV of the piglet birthweight within the litter and piglet body weight at 1 day did not differ between the two systems (Table 3). However, at 1 day old, the piglets raised in the free-farrowing system had gained more weight than those in the crate system (*p* = 0.012, Table 3). Piglets born in the free-farrowing system ingested more colostrum than those in the crate system (*p* = 0.019, Figure 2b). The number of piglets at weaning tended to be higher in sows kept in the crate system than in those kept in the free-farrowing system (*p* = 0.080, Table 3). Similarly, the litter weight of piglets at weaning for the sows kept in the crate system also tended to be higher than that of piglets from sows kept in the free-farrowing system (*p* = 0.078, Table 3).

#### 3.2.2. Piglet Preweaning Mortality

The total piglet preweaning mortality rate and the mortality rate classified by causes in the free-farrowing system compared with the crate system are presented in Table 4. Interestingly, the piglet preweaning mortality rate (26.8 ± 2.9% vs. 17.0 ± 3.8%, *p* = 0.045) and the proportion of piglets crushed by sows (13.1 ± 2.1% vs. 5.8 ± 2.7%, *p* = 0.037, Figure 3) were higher in the free-farrowing than in the crate system. The proportion of piglets crushed by sows did not differ between the two farrowing systems during the first 3 days postpartum (*p* > 0.05), but a difference was observed after 4 to 21 days of lactation (*p* = 0.008, Table 4). In the free-farrowing system, the piglet preweaning mortality rate in sows with high backfat thickness was higher than that in sows with moderate (37.8 ± 5.1% vs. 21.6 ± 3.6%, *p* = 0.011) and low (21.0 ± 6.2%, *p* = 0.038) backfat thickness. In addition, piglet preweaning mortality for high-backfat sows in the free-farrowing system was greater than that for crated sows at both 3 days (17.9 ± 3.9% vs. 4.8 ± 5.8%, *p* = 0.065) and 21 days of age (37.8 ± 5.1% vs. 10.9 ± 8.0%, *p* = 0.006). Similarly, the incidence of crushing in sows with high backfat thickness was higher in the free-farrowing system than in the crate system at both 3 days (11.1 ± 3.3% vs. 2.6 ± 4.9%, *p* = 0.055, Figure 4a) and 21 days of age (17.6 ± 3.6% vs. 4.0 ± 5.7%, *p* = 0.049, Figure 4b).

## 4. Discussion

In the present study, the farrowing performance of sows, newborn piglet characteristics, colostrum yield, milk yield and piglet preweaning mortality were compared between two different farrowing systems, i.e., crate vs. free-farrowing pen, within the same herd and in the same farrowing house. The sows were under moderate heat stress because the average 24 h indoor temperature and humidity during the experimental period were 28.1 ± 1.5 °C and 74 ± 5.4%, respectively. Moreover, the proportion of days when the average temperature inside the barn rose above 25.0 °C was 97.4%. A previous study has demonstrated that the sow thermal preference during the late gestation period was only 14.0 °C [29], which is much lower than that observed in the present study. In general, heat stress in sows can occur when the ambient temperatures rises above 25 °C. This is one of the major problems that decreases daily feed intake and compromises the milk yield of sows under tropical conditions [30]. Furthermore, sow reproductive performance under tropical conditions can be compromised due to the effect of heat stress on the intestinal barrier function, which can limit digestive ability and allow potential pathogens and/or toxins to become systemic [30]. Therefore, all farrowing performance and piglet characteristics demonstrated herein represented those of sows kept in a tropical environment, different from previous studies in temperate areas [6,10,15,31,32,33]. Moreover, the free-farrowing system has recently been introduced to the Thai swine industry, and scientific data concerning the advantages and disadvantages of this new farrowing system are insufficient. The differences in both farrowing performance and piglet characteristics from birth until weaning between the free-farrowing system and the crate system are presented below.

### 4.1. Colostrum and Milk Yield

The colostrum yield of sows in the free-farrowing system was higher than that of sows in the crate system. Interestingly, the sows in the free-farrowing system produced 0.5 kg more colostrum than those in the crate system. Oxytocin plays a crucial role as the mediator for mammary myoepithelial cell contraction [34], and an increase in oxytocin around parturition is important for both colostrum production and secretion [34]. Oliviero et al. [6] demonstrated that the levels of oxytocin during farrowing in sows kept in pens were significantly higher than those of sows kept in crate systems. Thus, sows kept in the farrowing pen had a shorter farrowing duration than those in the crate system [6]. Moreover, Yun et al. [33] found that the concentration of prepartum plasma oxytocin of sows in the free-farrowing system with provision of nesting materials was 26.4% higher than that of sows in the crate system. These studies indicate that an increase in oxytocin concentration during pre- and peri-partum periods may attribute to a higher colostrum yield in sows kept in the free-farrowing system compared to the crate system. In the present study, exogenous oxytocin was frequently used in either the crated or the free-farrowing systems. The use of exogenous oxytocin during the peripartum period could be an important factor that influences the colostrum consumption of piglets. Previous studies have demonstrated that exogenous oxytocin administration can increase the number of stillborn and number of live-born piglets with ruptured umbilical cord, meconium staining and neonatal asphyxia [35,36]. These characteristics can influence piglet vitality and hence compromise their colostrum consumption ability. However, in the present study, the proportion of stillborn and meconium-stained piglets did not differ significantly between the crated and the free-farrowing systems.

In the present study, the average milk yield of sows in the free-farrowing system did not differ significantly compared to that of sows kept in the crate system. This indicates that the free-farrowing system has no deleterious effect on sow milk yield. Regardless of the farrowing system, the average milk yield of sows between 3 and 10 days and 10 and 17 days of lactation were 8.6 and 10.4 kg/day, respectively. These values are lower than those reported in an earlier study under tropical conditions, i.e., 10.4 and 12.8 kg/day, respectively [13]. In a previous study in Denmark, the average milk yield of sows at lactation peak was 9.23 kg [27]. In addition, backfat thickness before parturition influences sow milk yield. In the previous study, the milk yield of sows between 3 and 10 days of lactation increased as backfat thickness before parturition increased [13]. However, in the present study, high-backfat sows in the free-farrowing system had a lower milk yield than high-backfat sows in the crate system. The reason could be because sows with high backfat thickness had more mammary parenchymal tissue and more total protein and total DNA than sows with moderate and low backfat thickness [37]. Therefore, increasing parenchymal tissue in late gestation is the major factor that enhances milk production and the growth of suckling piglets [37]. Another reason could be due to a higher piglet mortality rate and a higher proportion of crushed piglets in the high-backfat sows kept in the free-farrowing system, with a consequent reduction in the number of suckling piglets. Therefore, the estimated milk yield was also reduced. Thus, if the number of crushed piglets in the high-backfat sows was reduced, the milk yield of sows in the free-farrowing system might have been increased.

### 4.2. Piglet Preweaning Mortality

The piglet preweaning mortality rate and the proportion of piglets crushed by sows in the free-farrowing system were 26.8% and 13.1%, respectively. On the other hand, the piglet preweaning mortality rate and the proportion of piglets crushed by sows in the crate system were only 17.0% and 5.8%, respectively. To our knowledge, the present study is the first study demonstrating the piglet preweaning mortality rate and the proportion of piglets crushed by sows in the free-farrowing system under a tropical climate. The average piglet preweaning mortality rate observed in the present study is relatively high but still within the normal range reported earlier in either the crate or the free-farrowing system [16,19,38,39,40,41,42]. In the conventional crate system, the piglet preweaning mortality rate in swine commercial herds in Thailand averages 11.2% and varies among herds from 4.8% to 19.2% [16]. In the free-farrowing system in European countries, the average piglet preweaning mortality rate ranges from 5.1% to 26.0% [19,38,39,40,41,42]. The relatively high piglet preweaning mortality observed in the present study could be related to heat stress in pre- and peri-partum sows because the average temperature inside the farrowing house was, in most cases, above 25.0 °C [43]. A recent study has demonstrated that the risk of piglet mortality in the free-farrowing system was 14% higher than that in the crate system [44]. The present study demonstrated that, within the same herd and the same management, the piglet preweaning mortality rate in the free-farrowing pen was 9.8% higher than that in the crate system. However, a study in Denmark found that the difference in piglet preweaning mortality between free-farrowing and crate systems was only 1.3%, i.e., 13.7% vs. 11.8%, respectively [41]. This indicates that the major disadvantage of the free-farrowing system is the risk of having a high piglet-preweaning mortality. However, the differences among studies indicate that the high piglet-preweaning mortality in the free-farrowing system is a manageable trait and could be overcome by improving various husbandry strategies. For instance, in a previous study, a temporary crate system during some periods of lactation was recommended [42]. However, the total piglet mortality in the temporary confinement system was only slightly decreased compared to that of the complete free-farrowing system, i.e., 25.4% vs. 26.0%, respectively [42]. This indicates that some underlying factors associated with piglet preweaning mortality in the free-farrowing system remain to be further elucidated.

Interestingly, the present study also demonstrated that the proportion of piglets crushed by sows in the free-farrowing pen was 7.3% higher than that in the crate system. This is in agreement with a number of previous studies in temperate areas [31,39,42,45]. For example, in China, the percentage of piglets crushed by sows in the farrowing pen was 14.7% higher than that in the crate system, i.e., 25.5% vs. 10.8%, respectively [45]. However, data regarding the proportion of piglets crushed by sows in the free-farrowing system in the tropics have never been reported. In Finland, the proportion of piglets crushed by sows in the farrowing pen was 14.2% greater than that in the farrowing crate [31]. In Germany, the proportion of piglets crushed by sows in the farrowing pen accounted for up to 70.8% of the piglet preweaning mortality [39]. Similarly, in Switzerland, crushing accounted for 53.4% of the piglet preweaning mortality in the free-farrowing pen [38]. In the present study, crushing by sows accounted for 48.9% of the total piglet preweaning mortality in the free-farrowing pen. On the other hand, Loftus et al. [40] recently demonstrated that the proportion of piglets crushed by sows in the free-farrowing system can be reduced to 3.5%, without a significant difference to the crate system (i.e., 3.3%), by using a large size of the farrowing pen (i.e., 5.6 m^2^). In the present study, the total area of the pen was 4.7 m^2^, and the space available for a sow in the farrowing pen was only 3.25 m^2^, much lower than that recently recommended by the European Food Safety Authority (EFSA), i.e., above 6.6 m^2^ for the complete free-farrowing system and 4.3 to 6.3 m^2^ for the temporary crating system [46]. Therefore, the incidence of crushing in the free-farrowing system observed in the present study was relatively high (13.1%) compared to that reported in UK (i.e., 3.5%) [40]. This indicates that some husbandry practices, as well as the equipment used in the farrowing house, may help in minimising the proportion of piglets crushed by sows in the free-farrowing system. Additionally, poor maternal behaviours, e.g., rapid postural change and rolling and stepping on the piglets, can also contribute to the high incidence of crushing in the free-farrowing pen [31,39]. Interestingly, the present study found that the proportion of piglets crushed by sows was higher for sows with high backfat thickness (i.e., >24 mm). This is in agreement with Rangstrup-Christensen et al. [47], who observed an increase in piglet preweaning mortality in sows that had a high body-condition score. Most likely, high-backfat sows frequently step on their piglets while lying down [48]. Moreover, these sows frequently change posture and spend more time standing, consequently trapping their piglets [31]. These data indicate that the high prevalence of piglets crushed by sows in the free-farrowing system is usually observed for sows with a relatively high backfat thickness. Thus, additional management strategies to avoid crushing by sows in the free-farrowing system should be focused on sows with high backfat thickness (i.e., >24 mm), trying to minimise the proportion of sows with high body condition before parturition.

### 4.3. Farrowing Performance and Piglet Characteristics

The farrowing duration of sows in the free-farrowing system did not differ significantly from that of sows in the crate system (199.3 vs. 213.3 min). This is in contrast to a previous study in Finland [15], wherein sows kept in pens had a shorter farrowing duration than those kept in crates, i.e., 212 vs. 301 min, respectively [15]. Furthermore, in a temperate area, Yun et al. [31] revealed that modern hyperprolific sows with an average of 19.3 piglets per litter and kept in a free-farrowing system had a much longer farrowing duration (i.e., 399.4 min) compared to those in the present study. This might be explained by the fact that the total number of piglets born per litter in the Finnish study was four piglets higher than that observed in the present study (15.3 vs. 19.3 piglets/litter) [31]. In the present study, 22.1% and 22.2% of sows in the crate system and the free-farrowing system, respectively, had a prolonged duration of farrowing (i.e., >4 h). Farrowing duration is strongly associated with the concentration of oxytocin, which plays a major role in farrowing progression as it binds to receptors in myometrial cells and stimulates calcium as a second messenger for contraction [8]. During farrowing, the oxytocin concentration was higher in penned sows compared to those kept in crates, i.e., 77.6 vs. 38.1 pg/mL, respectively [6]. Moreover, Blim et al. [49] found that the total calcium concentration in serum at the beginning of the expulsion stage was higher in penned sows compared to crated sows [49]. Thus, if the litter size at birth in the free-farrowing sows under tropical conditions is increased, farrowing duration may also be increased, and the benefits of this farrowing system may be detected.

In the present study, the incidence of stillborn piglets did not differ between the two farrowing systems. This finding agrees with previous studies in temperate environments [15,31]. In additions, the present study is the first that demonstrates the incidence of piglets born with meconium staining (46.6%) and IUGR piglets (13.2%) in the free-farrowing system in a tropical environment. In piglets, meconium staining is associated with umbilical cord rupture and asphyxia [50,51]. Nevertheless, the incidence of either piglets born with meconium staining or IUGR characteristics observed in the present study did not differ between the pen and the crate systems. This indicates that piglets from both systems experienced similar levels of growth retardation and asphyxia. Therefore, the free-farrowing system in tropical environments has no negative impact on newborn piglet characteristics.

### 4.4. Sow Backfat Thickness and Loin Muscle Depth

Sow backfat thickness and loin muscle depth before parturition and 21 days of lactation and the relative backfat and loin muscle loss during lactation did not significantly differ between sows kept in the farrowing crate and the free-farrowing pen. This finding is in line with the results of a previous study conducted in Europe, which proposed no effects of the farrowing pen on lactational body weight and backfat loss in sows [52]. This indicates a similar backfat and loin muscle loss during lactation in the two different farrowing systems under tropical conditions. Regardless of the farrowing system, lactational backfat loss was higher in sows with high backfat thickness before farrowing compared to those with low and moderate backfat thickness. This is in agreement with a previous study in Thailand [13]. However, the average backfat thickness in the previous study [13] was lower compared to that of the present study. Additionally, our previous study demonstrated that the percentage of sows losing backfat >10% during lactation was higher when backfat was >25.0 mm before farrowing (85.7%) compared to backfat levels of 15.0 to 20.0 mm before farrowing (35.0%) [53]. The difference in sow backfat thickness observed among these studies might be due to the different genetic lines. Therefore, the optimal backfat thickness of sows can vary among herds and genetics lines.

## 5. Conclusions

Gestation length, stillbirth, farrowing duration, piglet expulsion interval and time from the onset of farrowing to the last placental expulsion in sows kept in farrowing crates did not differ significantly compared to those in sows kept in free-farrowing system. Piglets born in the free-farrowing system had a higher colostrum intake than those in the crate system. However, the piglet preweaning mortality rate and the proportion of piglets crushed by sows in the free-farrowing pen were higher than those in the crate system. Interestingly, a high proportion of piglet preweaning mortality in the free-farrowing pen was detected only in sows with high backfat thickness (>24 mm) before farrowing but not in those with low and moderate backfat thickness. Therefore, special attention, e.g., temporary confinement, can be recommended for sows with high backfat thickness to avoid the crushing of piglets.

## Figures and Tables

**Figure 1 animals-13-00233-f001:**
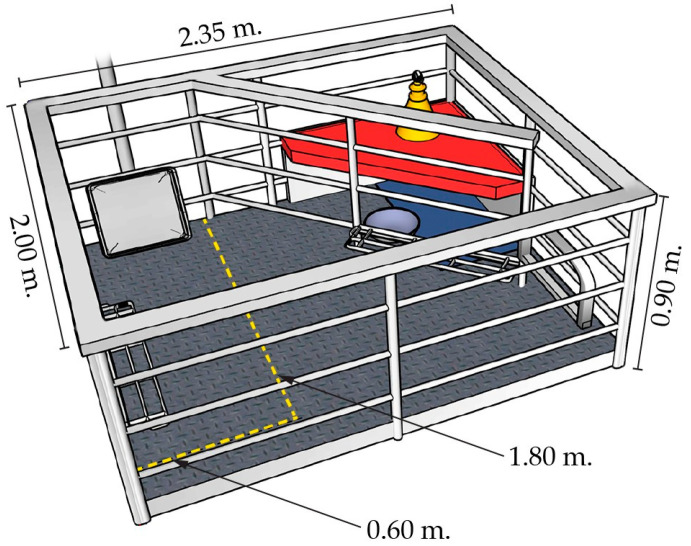
Schematic diagram of the farrowing pen with a lockable swing hinge providing the total area of 4.7 m^2^ per pen (2.0 × 2.35 × 0.9 m). In the crate system, the swing hinge was closed (yellow dashed line) from entering the farrowing unit until weaning (sow space allowance = 1.08 m^2^). In the free-farrowing system, the hinge was completely opened and locked with one side of the pen. The pen remained opened from entering to the farrowing unit until weaning (sow space allowance = 3.25 m^2^). The feeding box was located in the front of the pen. The creeping area was covered with a red plastic roof and consisted of a heating lamp, a rubber mattress and a piglet feeding bowl.

**Figure 2 animals-13-00233-f002:**
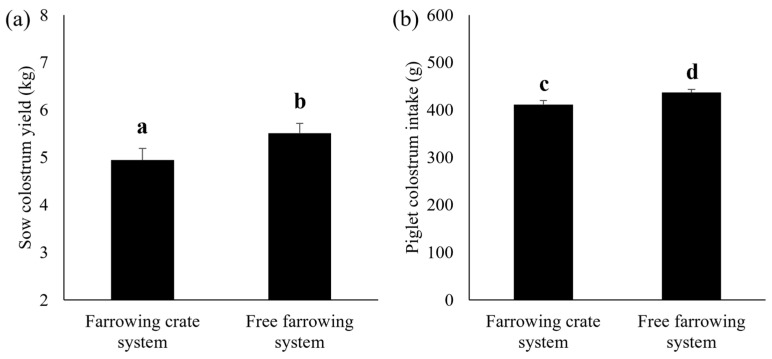
(**a**) Sow colostrum yield and (**b**) piglet colostrum intake for sows kept in the crate system and the free-farrowing system. Data are presented as Lsmeans and SEM. a and b superscripts indicate a tendential difference (*p* = 0.080). c and d superscripts indicate a significant difference (*p* < 0.05).

**Figure 3 animals-13-00233-f003:**
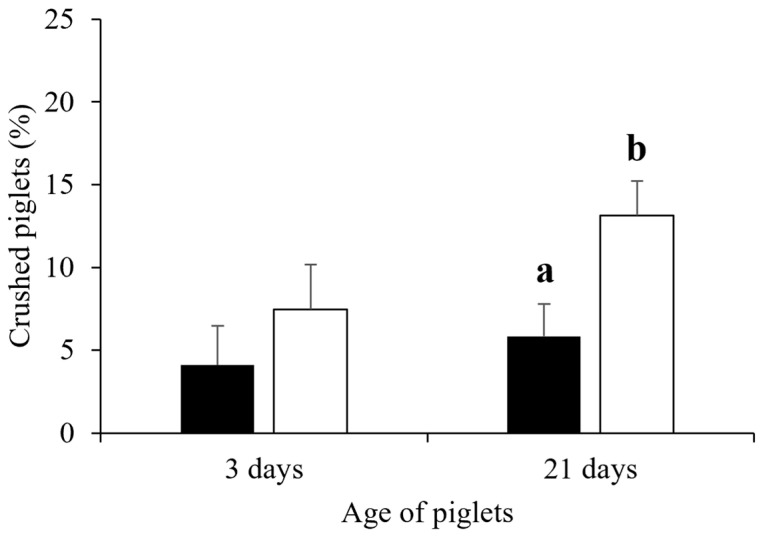
Proportion of piglets dead due to crushing by sows from litters raised either in the crate system (black bar) or the free-farrowing system (white bar) between 0 and 3 days of age and 0 to 21 days of age. Data are presented as Lsmeans and SEM. a and b superscripts indicate a significant difference (*p* < 0.05).

**Figure 4 animals-13-00233-f004:**
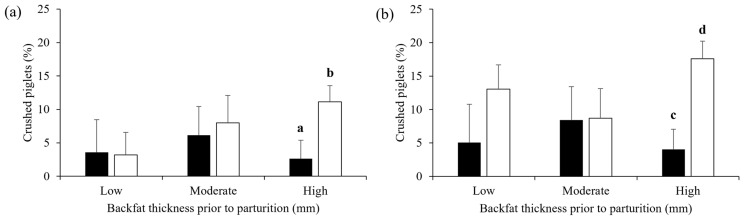
Preweaning mortality of piglets from litters raised either in the crate system (black bar) or the free-farrowing system (white bar) from sows of three different backfat classes before farrowing, low (<18 mm), moderate (18 to 24 mm) and high (>24 mm). (**a**) Between 0 to 3 days of age and (**b**) between 0 to 21 days of age. Data are presented as Lsmeans and SEM. a and b superscripts indicate a tendential difference (*p* = 0.055). c and d superscripts indicate a significant difference (*p* < 0.05).

**Table 1 animals-13-00233-t001:** Gestation length, litter traits and metabolic parameters in sows kept in the crate system compared to sows kept in the free-farrowing system in a tropical environment (Lsmeans ± SEM).

Variables	Crate System	Free-Farrowing System	*p* Value
Number of sows	38	54	
Parity number ^1^	2.0 ± 0.5	2.2 ± 0.6	
Gestation length (d)	114.4 ± 0.3	114.8 ± 0.2	0.320
Total number of piglets born per litter	14.7 ± 0.5	15.3 ± 0.4	0.365
Number of piglets born alive per litter	12.9 ± 0.6	13.2 ± 0.5	0.699
Stillborn piglets per litter (%)	9.8	8.9	0.757
Mummified foetuses per litter (%)	2.1	4.8	0.191
Backfat thickness prior to parturition (mm)	20.7 ± 0.6	21.2 ± 0.5	0.549
Loin muscle depth prior to parturition (mm)	48.2 ± 0.7	49.4 ± 0.6	0.202
Backfat thickness at 21 days of lactation (mm)	15.1 ± 0.5	15.0 ± 0.4	0.907
Loin muscle depth 21 days of lactation (mm)	41.5 ± 0.8	41.2 ± 0.6	0.822
Lactational backfat loss (%)	25.1	28.3	0.288
Lactational loin muscle loss (%)	14.0	15.8	0.445
Sow loss backfat during lactation >20% (%)	62.9	75.5	0.204
Sow loss loin muscle during lactation >10% (%)	62.9	69.8	0.497
Weaning-to-service interval (days)	4.5	5.1	0.435

^1^ Means ± SD.

**Table 2 animals-13-00233-t002:** Gestation length, litter traits and metabolic parameters of sows with low (<18 mm), moderate (18 to 24 mm) and high (>24 mm) backfat thickness prior to parturition (Lsmeans ± SEM).

Variables	Backfat Thickness Prior to Parturition (mm)		
	Low	Moderate	High
Number of sows	19	51	22
Parity number ^1^	2.2 ± 0.5	2.0 ± 0.5	2.2 ± 0.7
Gestation length (d)	114.3 ± 0.4	114.8 ± 0.2	114.6 ± 0.4
Farrowing duration (min)	176.9 ± 33.6	208.9 ± 20.5	258.2 ± 32.4
Total number of piglets born per litter	15.7 ± 0.6 ^a^	14.0 ± 0.4 ^b^	15.2 ± 0.6 ^ab^
Number of born alive piglets per litter	13.6 ± 0.7	12.2 ± 0.4	13.2 ± 0.7
Stillborn piglets per litter (%)	10.0	8.6	9.4
Mummified foetuses per litter (%)	3.1	3.5	3.7
Backfat thickness prior to parturition (mm)	15.1 ± 0.4 ^a^	21.2 ± 0.3 ^b^	25.7 ± 0.4 ^c^
Loin muscle depth prior to parturition (mm)	46.3 ± 0.9 ^a^	48.4 ± 0.6 ^a^	52.2 ± 0.9 ^b^
Backfat thickness at 21 days of lactation (mm)	12.5 ± 0.7 ^a^	15.2 ± 0.4 ^b^	17.1 ± 0.7 ^c^
Loin muscle depth 21 days of lactation (mm)	39.3 ± 1.0 ^a^	41.4 ± 0.6 ^ab^	43.6 ± 1.1 ^b^
Lactational backfat loss (%)	16.5 ± 3.1 ^a^	28.3 ± 1.9 ^b^	33.4 ± 3.2 ^b^
Lactational loin muscle loss (%)	15.0 ± 2.6	14.3 ± 1.6	16.6 ± 2.7
Weaning-to-service interval (days)	4.9 ± 0.3	4.6 ± 0.7	5.0 ± 1.0

^1^ Means ± SD. a, b and c superscripts indicate statistical significance (*p* < 0.05).

**Table 3 animals-13-00233-t003:** Farrowing performance and piglet characteristics in the litters in the crate system compared to the litters in the free-farrowing system in a tropical environment (Lsmeans ± SEM).

Variables	Crate System	Free-Farrowing System	*p* Value
Number of sows	38	54	
Farrowing duration (min)	229.9 ± 26.5	199.3 ± 21.3	0.371
Time from onset of farrowing to the last placental expulsion (min)	471.5 ± 51.2	384.4 ± 42.8	0.196
Proportion of sow farrowed longer than 240 min (%)	21.1	22.2	0.894
Coefficient of variance of piglet birthweight (%)	21.4	21.3	0.974
Number of piglets	539	805	
Piglet expulsion interval (min)	12.9 ± 0.8	12.4 ± 0.7	0.612
Cumulative expulsion interval (min)	101.8 ± 4.3	96.7 ± 3.5	0.361
Individual birthweight (g)	1297 ± 15	1308 ± 12	0.570
Proportion of piglets with body weight <1.0 kg (%)	16.2	18.6	0.281
Individual piglet body weight at 1 day old (g)	1388 ± 17	1420 ± 14	0.142
Body weight gain during the first 24 h (g)	85.3 ± 6.2	105.5 ± 5.1	0.012
Meconium-stained piglets (%)	49.4	46.6	0.347
IUGR piglets (%) ^1^	13.6	13.2	0.830
Number of piglets at weaning per litter	11.0 ± 0.5	10.0 ± 0.3	0.080
Litter weight at weaning (kg)	55.3 ± 2.7	49.1 ± 2.2	0.078

^1^ Intrauterine growth restriction.

**Table 4 animals-13-00233-t004:** Causes of piglet mortality during the lactation period in the crate system and the free-farrowing system.

Causes of Piglet Mortality	Crate System	Free-Farrowing System	*p* Value
All piglet preweaning mortality (*n* = 260)			
- Total mortality (%)	17.0 ± 3.8	26.8 ± 2.9	0.045
- 0 to 3 days of age (%)	10.9 ± 2.8	12.6 ± 2.3	0.628
- 4 to 21 days of age (%)	5.3 ± 2.1	14.2 ± 1.6	0.001
Crushing by sow (*n* = 104)			
- Total mortality (%)	5.8 ± 2.7	13.1 ± 2.1	0.037
- 0 to 3 days of age (%)	4.1 ± 2.4	7.4 ± 1.9	0.279
- 4 to 21 days of age (%)	1.4 ± 1.2	5.7 ± 1.0	0.008
Weak (*n* = 121)			
- Total mortality (%)	17.0 ± 3.8	26.8 ± 2.9	0.050
- 0 to 3 days of age (%)	6.1 ± 1.5	4.8 ± 1.2	0.501
- 4 to 21 days of age (%)	3.0 ± 1.3	5.7 ± 1.0	0.116
Miscellaneous causes (*n* = 35)			
- Total mortality (%)	1.4 ± 1.6	3.2 ± 1.2	0.376
- 0 to 3 days of age (%)	0.7 ± 0.5	0.5 ± 0.4	0.743
- 4 to 21 days of age (%)	0.6 ± 1.3	2.7 ± 1.0	0.197

## Data Availability

The data presented in this study are available on request from the corresponding author.

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
