# Peer review of "Pen Versus Crate: A Comparative Study on the Effects of Different Farrowing Systems on Farrowing Performance, Colostrum Yield and Piglet Preweaning Mortality in Sows under Tropical Conditions"

_animals, 2023, doi:10.3390/ani13020233_

Round 1

Reviewer 1 Report

The present study uses a topic that is highly relevant in the porcine industry and animal welfare. Free-farrowing systems are the alternative to conventional management of sows, improving their welfare and reducing the stress response that confinement can cause in animals. However, this systems need to be studied in depth before proposing to change all porcine farms since some of the obtained results in the present study need to be considered. Therefore, this paper is well organized and provides novel information that will be helpful for future research, particularly in tropical areas dedicated to pig farm. I made some minor comments hoping they can be useful for the authors.

Lines 66-67: I recommend adding a couple of reasons why gestation crates are prohibited. This is associated to a poor environment that does not met biological need of the animals (this article might be helpful: https://doi.org/10.3390/ani12070928). That would give the reader a claer idea why free-farrowing systems are so important but need to be studied.

Response:

Lines 81-82: Here as well. It might be appropriate to include a couple of reasons why piglet mortality occurs in the first 72 h.

Response:

Lines 92-96: Consider moving this paragraph after line 117, and before the sentence “To our knowledge….”. That way you can give continuity to the idea that the parameters you evaluated haven’t been studied before.

Response:

Lines 107-109: I would remove these lines since they might sound a little repetitive. The recommendation to put at the end of the introduction that this issue hasn’t been studied in tThe tropics would help to compile all the parameters and highlight the novelty of your study.

Response:

Line 152: if this value is the recommended allowance space in Thailand, it would help to mention it or add a reference.

Response:

Line 154: Is there any previous study where this space allowance was used? Or how did you decide on this size? If there is an article, I would recommend citing it.

Response:

Line 157: Please, add the brand, company name, and country of the lactation diet.

Response:

Line 166-170: Please, include the date when these procedures were performed, or if they were performed all at once the same day.

Response:

Table 4: Revise the style of the line “Miscellaneous causes". Maybe it needs to be at the same level as weak, crushing, and all piglets?

Response:

Lines 411-416: Do the authors consider that the moderate heat stress in the sows could have affected the assessed parameters? If so, this could be a limitation of the study that could be mentioned.

Response:

Lines 428-438: It would be interesting to also discuss the influence that oxytocin administration in penned, crated or free-farrowing system has on the sow and the fetus since oxytocin was used in the present study and is an important factor that needs to be mentioned. Previous authors have reported that exogenous oxytocin administration can increase the number of stillborn with ruptured umbilical cord, meconium staining, and fetal stress signs such as neonatal asphyxia. I recommend to check this articles among others: https://doi.org/10.1016/j.anireprosci.2003.11.002 and https://doi.org/10.1016/j.anireprosci.2005.04.012

Response:

Lines 446-447: How does back fat thickness associate with milk yield? A brief description of this can be added, perhaps by the endocrine response, etc.

Response:

Reference list: Include the doi of all the articles with available DOI identifiers.

Response:

Author Response

#Reviewer 1

Comments and Suggestions for Authors

The present study uses a topic that is highly relevant in the porcine industry and animal welfare. Free-farrowing systems are the alternative to conventional management of sows, improving their welfare and reducing the stress response that confinement can cause in animals. However, this system needs to be studied in depth before proposing to change all porcine farms since some of the obtained results in the present study need to be considered. Therefore, this paper is well organized and provides novel information that will be helpful for future research, particularly in tropical areas dedicated to pig farm. I made some minor comments hoping they can be useful for the authors.

Lines 66-67: I recommend adding a couple of reasons why gestation crates are prohibited. This is associated to a poor environment that does not meet biological need of the animals (this article might be helpful: https://doi.org/10.3390/ani12070928). That would give the reader a clear idea why free-farrowing systems are so important but need to be studied.

Response: Additional information has been added as suggested: “The gestation crate fails to meet all of a sow’s biological requirements in part by limiting her ability to perform several natural behaviours including simply turning around. These altered behavioural responses result from central nervous system processing of both internal and external stimuli and can frustrate a sow, evoking negative emotional responses and potentially compromising her well-being [Coria-Avila et al., 2022: Animals 12, 928]. The limitation on behav-ioural response of sows prior to farrowing is associated with either physiological or endocrine systems during parturition and lactation periods. The behavioural response of sows is coordinated by the central nervous system from the processing of internal and external stimuli, facilitated sow biological requirements and well-being [Coria-Avila et al., 2022: Animals 12, 928].”

Lines 81-82: Here as well. It might be appropriate to include a couple of reasons why piglet mortality occurs in the first 72 h.

Response: Additional information has been added: “Even a short period of peri-parturient asphyxia and hypoxia can lead to brain damage, increase the piglet’s risk of being crushed by a sow, and compromise piglet vitality during early postnatal life [Alonso-Spilsbury et al., 2005: Anim Reprod Sci 90, 1–30].”

Lines 92-96: Consider moving this paragraph after line 117, and before the sentence “To our knowledge….”. That way you can give continuity to the idea that the parameters you evaluated haven’t been studied before.

Response: Modified as suggested. Also, introduction part has been re-arranged according to comments made by reviewer #2.

Lines 107-109: I would remove these lines since they might sound a little repetitive. The recommendation to put at the end of the introduction that this issue hasn’t been studied in the tropics would help to compile all the parameters and highlight the novelty of your study.

Response: Removed as suggested.

Line 152: if this value is the recommended allowance space in Thailand, it would help to mention it or add a reference.

Response: The space allowance is referred to the instruction of standard housing design in a private breeding company in Thailand. This space has been used recently and a reference using the same housing has been added [Adi et al., 2022: Animals 12, 2943]. However, the previous reference did not provide details in the farrowing house design but rather focus on the farrowing performance of sows in this housing system. In addition, the space allowance for sows in the free-farrowing system in Thailand has been designed following the criteria of the minimum space requirement for sows that are able to turn-around nest space for piglet inspection and gathering behaviour (i.e., 3.17 m2) in the farrowing pen, recommended by a previous study in UK [Baxter et al., 2011: Animal 5, 580–600; Baxter et al., 2022: Front Vet Sci. 14:998192]. These references have also been added. Also, this additional information has also been addressed in the Materials & Methods.

Line 154: Is there any previous study where this space allowance was used? Or how did you decide on this size? If there is an article, I would recommend citing it.

Response: Previous study recommended this space allowance has been added [Baxter et al., 2011: Animal 5, 580–600] and also previous study using the same space allowance in Thailand has also been mentioned [Adi et al., 2022: Animals 12, 2943].

Line 157: Please, add the brand, company name, and country of the lactation diet.

Response: The brand, company name, city and country has been added: “(907 BTG, Betagro Public Co. Ltd., Lopburi, Thailand)”.

Line 166-170: Please, include the date when these procedures were performed, or if they were performed all at once the same day.

Response: The date when the procedures were performed has been added: “Piglet general husbandry included iron injection (200 mg/piglet of Iron dextran, Bezter Irondex 100®, Thainaoka Pharmaceutical Co., Ltd., Samut Sakhon, Thailand) and teeth clipping were carried out at 1 day of age. Additionally, antiprotozoal drug provision (20 mg/kg of 5% Toltrazuril, Better Pharma Co., Ltd., Lopburi, Thailand) and castration were performed at 3 days of age.”

Table 4: Revise the style of the line “Miscellaneous causes". Maybe it needs to be at the same level as weak, crushing, and all piglets?

Response: Modified as suggested. Thank you.

Lines 411-416: Do the authors consider that the moderate heat stress in the sows could have affected the assessed parameters? If so, this could be a limitation of the study that could be mentioned.

Response: Yes, we consider that the moderate heat stress can be one of the factor affecting the assessed parameters. Additional discussion and reference concerning this issue have been addressed: “In general, heat stress in sows can occur when the ambient temperatures rise above 25°C. This is one of the major problems that decreases daily feed intake and compromises milk yield of sows under tropical conditions [Tummaruk et al., 2022: Mol Reprod Dev, 1 – 13]. Further-more, the sow reproductive performances under tropical conditions can be compromised due to the effect of heat stress on the intestinal barrier function, which can limit digestive ability and allow potential pathogens and/or toxins to become systemic [Tummaruk et al., 2022: Mol Reprod Dev, 1 – 13].”

Lines 428-438: It would be interesting to also discuss the influence that oxytocin administration in penned, crated or free-farrowing system has on the sow and the fetus since oxytocin was used in the present study and is an important factor that needs to be mentioned. Previous authors have reported that exogenous oxytocin administration can increase the number of stillborn with ruptured umbilical cord, meconium staining, and fetal stress signs such as neonatal asphyxia. I recommend to check this articles among others: https://doi.org/10.1016/j.anireprosci.2003.11.002 and https://doi.org/10.1016/j.anireprosci.2005.04.012

Response: Additional discussion and references have been added: “In the present study, exogenous oxytocin was frequently used in either the crated or the free-farrowing systems. The use of exogenous oxytocin during peripartum period could be an important factor that influence the colostrum consumption of piglets. Previous studies have demonstrated that exogenous oxytocin administration can increase the number of stillborn and number of live-born piglets with ruptured umbilical cord, meconium staining, and neonatal asphyxia [Alonso-Spilsbury et al., 2004: Anim Reprod Sci. 84, 157–167; Mota-Rojas et al., 2006: Anim Reprod Sci 92, 123–143]. These characteristics can influence the piglet vitality and hence compromise their colostrum consumption ability. However, in the present study, the proportion of stillborn and meconium-stained piglets did not differ significantly between the crated and the free-farrowing systems.”.

Lines 446-447: How does back fat thickness associate with milk yield? A brief description of this can be added, perhaps by the endocrine response, etc.

Response: Additional description and reference have been added: “The reason could be due to that sows with high backfat thickness had more mammary parenchymal tissue and more total protein and total DNA than sows with moderate and low backfat thickness [Farmer et al., 2017: Transl Anim Sci 1, 154–159]. Therefore, increasing parenchymal tissue in late gestation is the major factor that enhance milk production and growth of suckling piglets [Farmer et al., 2017: Transl Anim Sci 1, 154–159].”

Reference list: Include the doi of all the articles with available DOI identifiers.

Response: DOI of all references has been added.

Reviewer 2 Report

The research entitled "Pen Versus Crate: A Comparative Study on the Effects of Different Farrowing Systems on Farrowing Performance, Colostrum Yield and Piglet Preweaning Mortality in Sows under Tropical Conditions" is undoubtedly interesting both for the geographical context (tropical) in which it was conducted, and for the results obtained with particular regard to the findings concerning the relationship between the adiposity of sows and mortality due to crushing of piglets.

The manuscript appears very carefully prepared. The results are correctly reported and widely discussed. The bibliography is relevant and up-to-date. The various parts of the manuscript support the conclusions. However, in the reviewer's opinion, some improvements could be made to improve the quality of the paper further.

GENERAL COMMENTS

Introduction: The introduction contains several well-documented statements, but the information is not adequately arranged (i.e. dispersed with abrupt changes from one concept to another), and the purpose(s) of the research should be expressed more clearly. For example, the text repeats twice that there are no data regarding the effects of the free farrowing system in pig farming in tropical regions (lines 92-96, 107-109), and the study aims to fill this gap (lines 119-122). At the same time, it is unclear whether (or not) studying the effects of adiposity in sows was one of the initial research goals. Therefore, I recommend revising the introduction avoiding repetitions, connecting better the different parts and concepts to make it easier to read, and aiming at the fundamental goals of the study.

Discussion: Concerning pre-weaning mortality, I think the problem of adapting the structures (i.e., the pen) should be more deeply discussed. Some studies indicate that structural adjustments of free-farrowing pens allow for a reduction in mortality, bringing it to acceptable levels. In addition, it should be underlined and reported that the space available to sows in this research was less than recently recommended by EFSA (https://www.efsa.europa.eu/en/efsajournal/pub/7421)

SPECIFIC COMMENTS:

1)      Lines 66-67: I don't think lines 66-67 are entirely correct. In fact, in Europe, gestation in an individual cage is not prohibited but is limited to the first month of gestation and the week before farrowing.

 2)      How was colostrum collected for Brix index determination?

 3)      In consideration of the presence of multiparous sows (lines 130-131), it would be appropriate to specify what type of farrowing structures (traditional cage or free farrowing pen) sows had previously experienced since this affects behaviour in subsequent lactations.

 4)      To better define the productive response of sows under different conditions, and in consideration of the observations made on backfat thickness, it would be interesting to mention, if data are available, the effects on weaning to oestrous interval

Author Response

#Reviewer 2

 Comments and Suggestions for Authors

The research entitled "Pen Versus Crate: A Comparative Study on the Effects of Different Farrowing Systems on Farrowing Performance, Colostrum Yield and Piglet Preweaning Mortality in Sows under Tropical Conditions" is undoubtedly interesting both for the geographical context (tropical) in which it was conducted, and for the results obtained with particular regard to the findings concerning the relationship between the adiposity of sows and mortality due to crushing of piglets.

The manuscript appears very carefully prepared. The results are correctly reported and widely discussed. The bibliography is relevant and up-to-date. The various parts of the manuscript support the conclusions. However, in the reviewer's opinion, some improvements could be made to improve the quality of the paper further.

Response: Thank you very much for your time and your positive attitude on our manuscript. The manuscript has been revised according to the reviewer’s comments point-by-point. List of the responses to the reviewer are presented below:

GENERAL COMMENTS

Introduction: The introduction contains several well-documented statements, but the information is not adequately arranged (i.e. dispersed with abrupt changes from one concept to another), and the purpose(s) of the research should be expressed more clearly. For example, the text repeats twice that there are no data regarding the effects of the free farrowing system in pig farming in tropical regions (lines 92-96, 107-109), and the study aims to fill this gap (lines 119-122). At the same time, it is unclear whether (or not) studying the effects of adiposity in sows was one of the initial research goals. Therefore, I recommend revising the introduction avoiding repetitions, connecting better the different parts and concepts to make it easier to read, and aiming at the fundamental goals of the study.

Response: Thank you very much for comments. Introduction part has been revised intensively according to the comments made by both reviewers. Repetition has been avoided. For instance, information in line 92-96 has been moved to the same paragraph as the objective and Line 107-109 has been deleted. Additional objective concerning the influence of sow backfat thickness has been addressed.

Discussion: Concerning pre-weaning mortality, I think the problem of adapting the structures (i.e., the pen) should be more deeply discussed. Some studies indicate that structural adjustments of free-farrowing pens allow for a reduction in mortality, bringing it to acceptable levels. In addition, it should be underlined and reported that the space available to sows in this research was less than recently recommended by EFSA (https://www.efsa.europa.eu/en/efsajournal/pub/7421)

Response: Thank you very much for the advice and reference. Additional discussion concerning this issue has been added. “In the present study, the total area of the pen was 4.7 m2 and the space available for sow in the farrowing pen was only 3.25 m2, much lower than that recently recommended by the European Food Safety Authority (EFSA), i.e., above 6.6 m2 for complete free-farrowing system and 4.3 to 6.3 m2 for temporary crating system [EFSA et al., 2022: EFSA J 20, 7421]. Therefore, the incidence of crushing in the free-farrowing system observed in the present study was relatively high (13.1%) compared to that reported in UK (i.e., 3.5%) [Loftus et al., 2020: Appl. Anim. Behav. Sci. 232, 105102].”

SPECIFIC COMMENTS:

1) Lines 66-67: I don't think lines 66-67 are entirely correct. In fact, in Europe, gestation in an individual cage is not prohibited but is limited to the first month of gestation and the week before farrowing.

Response: The sentence has been changed to be: “….. the use of gestation crates has been limited or prohibited in most periods of pregnancy except for the first month of gestation and the week before farrowing [Einarsson et al., 2014: Acta Vet Scand 56, 37].”

2) How was colostrum collected for Brix index determination?

Response: Additional detail concerning colostrum samplings and brix index determination have been added in M&M: “The colostrum sample (0.3 mL) was collected manually from the first three pair of teats of the sows and was dropped into the prism chamber of the Brix refractometer using a disposable plastic dropper. The Brix index value was determined immediately after testing.”

3) In consideration of the presence of multiparous sows (lines 130-131), it would be appropriate to specify what type of farrowing structures (traditional cage or free farrowing pen) sows had previously experienced since this affects behaviour in subsequent lactations.

Response: Additional statement has been added in M&M: “For multiparous sows, the type of the farrowing structures that the sows had previously experienced was the conventional crate system.”

4) To better define the productive response of sows under different conditions, and in consideration of the observations made on backfat thickness, it would be interesting to mention, if data are available, the effects on weaning to oestrous interval.

Response: The data on weaning-to-service interval has been added in Table 1 and 2. Also this trait has been added to the statistical analysis and data collection part. Thank you very much.
